# Modeling and Analysis of a Simple Flexible Wing—Thorax System in Flapping-Wing Insects

**DOI:** 10.3390/biomimetics7040207

**Published:** 2022-11-21

**Authors:** Braden Cote, Samuel Weston, Mark Jankauski

**Affiliations:** Mechanical & Industrial Engineering, Montana State University, 220 Roberts Hall, Bozeman, MT 59717, USA

**Keywords:** insect flight, flapping-wing micro air vehicles, multi-body dynamics, structural mechanics, aerodynamics

## Abstract

Small-scale flapping-wing micro air vehicles (FWMAVs) are an emerging robotic technology with many applications in areas including infrastructure monitoring and remote sensing. However, challenges such as inefficient energetics and decreased payload capacity preclude the useful implementation of FWMAVs. Insects serve as inspiration to FWMAV design owing to their energy efficiency, maneuverability, and capacity to hover. Still, the biomechanics of insects remain challenging to model, thereby limiting the translational design insights we can gather from their flight. In particular, it is not well-understood how wing flexibility impacts the energy requirements of flapping flight. In this work, we developed a simple model of an insect drive train consisting of a compliant thorax coupled to a flexible wing flapping with single-degree-of-freedom rotation in a fluid environment. We applied this model to quantify the energy required to actuate a flapping wing system with parameters based off a hawkmoth *Manduca sexta*. Despite its simplifications, the model predicts thorax displacement, wingtip deflection and peak aerodynamic force in proximity to what has been measured experimentally in flying moths. We found a flapping system with flexible wings requires 20% less energy than a flapping system with rigid wings while maintaining similar aerodynamic performance. Passive wing deformation increases the effective angle of rotation of the flexible wing, thereby reducing the maximum rotation angle at the base of the wing. We investigated the sensitivity of these results to parameter deviations and found that the energetic savings conferred by the flexible wing are robust over a wide range of parameters.

## 1. Introduction

Micro air vehicles (MAVs) have become a ubiquitous technology over the past several years. They have been used in a variety of applications, including search and rescue [1], environmental sensing [2], and infrastructure monitoring [3]. Recently, there have been efforts to reduce the length scale of MAVs to expand the environments they can operate in. Centimeter- or millimeter-scale MAVs, for example, may be employed in environments such as congested piping networks or dense forestry that may be difficult for larger aircraft to navigate. However, the efficiency of conventional rotary or fixed-wing robotic vehicles is challenged at small-length scales (and hence, a low Reynolds numbers) because lift-generating aerodynamic forces are on the same order as viscous forces [4]. Many MAVs consequently use flapping wings, such as those employed by flying insects, to realize flight at small-length scales [5]. Nonetheless, the reduced payload capacity available to flapping-wing micro air vehicles (FWMAVs) introduces challenging design constraints. Some FWMAVs rely on tethers to provide power because drivetrain systems consume more power than onboard batteries are capable of producing [4]. Tethering limits the autonomy of the robotic vehicle and the applications where it is useful. By contrast, insects are extremely energy-efficient, sometimes sustaining autonomous flight for hours at a time [6]. Thus, a detailed understanding of the insect flight system can help overcome some of the current FWMAV limitations and lead to bio-inspired design guidelines.

Most flying insects enhance the energetic economy of flapping flight through “indirect actuation”, where the insect’s flight muscles attach to the thoracic exoskeleton rather than the wing base [7]. Two antagonistic sets of indirect flight muscles (IFMs) deform the thorax, and small thorax deformation is amplified into large wing rotation via a sophisticated linkage system called the wing hinge [8]. The flexible thorax stores and releases potential energy throughout a flapping cycle, thereby reducing the power invested to decelerate the wing upon stroke reversal [9]. Further, numerous insects are believed to flap at the resonant frequency of their thorax–wing system, thereby further exploiting the energetic savings associated with resonance [10]. When flapping at resonance, the IFMs require lower forces to deform the thorax compared to if the thorax were compressed slowly. The thorax is, therefore, a critical energy-saving structure in flapping wing insect flight.

In addition to the thorax, the insect’s flexible wings contribute to flight performance efficiency as well. Insect wings contain no musculature and thus, the insect has little control over the instantaneous shape of the wing during flight [11]. Instead, the bending and twisting experienced by flexible wings occurs passively under inertial and aerodynamic forces [12,13]. Wing deformation is believed to benefit flight in several ways. Experimental studies show that when the bumble bee wings are splinted to inhibit natural deformation, the insect suffers an approximate 9% loss in maximum achievable vertical force [14]. Further, computational models indicate that wing deformation in hawkmoths increases the wing’s lift-to-drag ratio by about 20–30% compared to that of a rigid wing at Reynolds numbers of about 75–1000 [15]. Wing deformation may also reduce the inertial power required to flap the wing [16], though an energy storage element upstream of the wing (such as the flexible thorax) may be required [17].

While many mathematical modeling efforts to date have focused on individual components (e.g., the isolated wing or thorax) rather than the system-level dynamics, some have formulated simple single-degree-of-freedom (SDOF) models to investigate coupling between the thorax, wing and surrounding fluid environment [18,19]. In the simplest configuration, a rigid wing and thorax are coupled via a static linear transmission, where a prescribed amount of thorax displacement corresponds to a certain amount of wing rotation. The aerodynamic forces acting on the wing are estimated via quasi-static airfoil theory and are propagated to the thorax through the wing–thorax transmission. Though SDOF models cannot capture the multiple-degree-of-freedom (MDOF) rotation of the wing (and hence cannot predict aerodynamic lift), they are useful when interrogating biophysical phenomena of the wing–thorax system. For example, an SDOF model and accompanying robophysical experiment showed that the resonant frequency of the wing–thorax system was dependent on wing rotation amplitude due to nonlinear quadratic fluid damping [18]; absent nonlinear damping, the resonant frequency depended only on the mass and stiffness of the wing and thorax. SDOF models have also demonstrated that flying insects can achieve energy savings over a broad frequency range, whereas it was previously believed that optimal energetic performance was centered narrowly around the system’s resonant frequency [19].

Despite insights garnered from these system-level models, it remains unknown how *flexible* wings contribute to the dynamics of the coupled wing–thorax system. To our knowledge, previous system-level models of the flight mechanism have considered only rigid wings. Prior work suggests that wing flexibility can reduce the energetic requirements of flapping only if there is an elastic storage element, such as the compliant thorax to brake the wing motion [17]. The objective of the current study is, therefore, to develop a simplified system-level model to investigate the dynamics of the flexible wing coupled to a compliant thorax and the surrounding fluid environment. While the derived model cannot describe aerodynamic lift, it provides a foundation to better assess energy flow through a multi-body system with fluid dissipation. Non-lifting models are simple yet useful tools for investigating the physics of flapping flight and are common in precedent work [18,19]. Insights garnered from this model can inform the compliant design of bio-inspired FWMAVs and lay a foundation for more complex models that accommodate MDOF rotations and lifting flows.

## 2. Mathematical Modeling

Here, we derive a simple model representative of the insect flight system (Figure 1). The model consists of a compliant thorax treated as a spring mass connected to two flexible wings modeled as elastic beams via a linear transmission. Wing deformation is determined via a weighted linear combination of the wing’s vibration mode shapes, and aerodynamic forces acting on the flapping wings are estimated using a modified blade element theory. We use the Lagrangian formulation to derive the coupled equations of motion governing the wing–thorax response and the principle of virtual work to incorporate non-conservative aerodynamic forces.

### 2.1. Kinematics

Consider the model schematic pictured in Figure 1. The thorax is treated as a spring mass damper with effective mass *m* and stiffness *k* and is constrained to move vertically with displacement *x*. The thorax is connected to two flexible wings via static transmissions. Assuming the transmissions behave linearly, the thorax displacement *x* is related to the wing’s rigid body rotation θ by
(1)x=Γθ
where Γ is the transmission ratio. This transmission implies that the thorax displacement and wing rigid body rotation maintain a fixed phase relationship over a wingbeat. Each wing rotates about a fixed point (pictured as *O* for the left wing in Figure 1). We establish a normal–tangential coordinate system that rotates with the rigid body motion of the wing, where ${\hat{e}}_{t}$ denotes a tangential unit vector and ${\hat{e}}_{n}$ denotes a normal unit vector. The wings experience elastic deformation W(y,t) with respect to the wing’s rigid body motion, where *y* denotes the position along the wing that exists between the wing root *O* and total wing length *L*, and *t* denotes time. The position ${\hat{r}}_{w}$ and velocity ${\hat{v}}_{w}$ at any point along the wing are
(2)$$\begin{array}{ccc} {\hat{r}}_{w} & = & y{\hat{e}}_{t}+W{\hat{e}}_{n} \end{array}$$
(3)$$\begin{array}{ccc} {\hat{v}}_{w} & = & -W\dot{\theta }{\hat{e}}_{t}+\left(y\dot{\theta }+\dot{W}\right){\hat{e}}_{n} \end{array}$$

Assuming that the wing deformation is small, it can be represented via an eigenfunction expansion such that
(4)$$W\left(y,t\right)=\sum_{k=1}^{\infty }\phi_{k}\left(y\right)q_{k}\left(t\right)$$
where $\phi_{k}$ is the wing’s *k*th vibration mode and $q_{k}$ is the time-dependent participation factor of $\phi_{k}$, also called the modal response. The mode shapes are normalized with respect to the wing mass $m_{w}$ and satisfy orthonormality such that
(5)$$\int_{m_{w}}\phi_{k}\phi_{r}=\delta_{kr}$$
where $\delta_{kr}$ is the Kronecker delta.

### 2.2. Equations of Motion

We use the energy-based Lagrangian approach to determine the coupled equations of motion governing the thorax and wing responses. For the thorax, the kinetic energy $T_{thorax}$ and potential energy $U_{thorax}$ are
(6)$$\begin{array}{ccc} T_{thorax} & = & \frac{1}{2}m{\dot{x}}^{2} \end{array}$$
(7)$$\begin{array}{ccc} U_{thorax} & = & \frac{1}{2}kx^{2} \end{array}$$

The kinetic energy of an individual wing, $T_{wing}$, is
(8)$$\begin{array}{ccc} T_{wing} & = & \frac{1}{2}\int_{m_{w}}{\hat{v}}_{w}\cdot {\hat{v}}_{w}\;dm_{w} \end{array}$$
(9)$$\begin{array}{ccc} T_{wing} & = & \frac{1}{2}\int_{m}\left[{\dot{\theta }}^{2}\left(y^{2}+W^{2}\right)+2\dot{\theta }y\dot{W}+{\dot{W}}^{2}\right]\;dm_{w} \end{array}$$

Expanding the above in terms of Equations (Equation 1) and (Equation 4) gives
(10)$$T_{wing}=\left[\frac{{\dot{x}}^{2}}{\Gamma^{2}}\left(I_{O}+\sum_{k=1}^{\infty }q_{k}^{2}\right)+\sum_{k=1}^{\infty }\left(2\frac{\dot{x}}{\Gamma }\lambda_{k}{\dot{q}}_{k}+{\dot{q}}_{k}^{2}\right)\right]$$
where $I_{O}$ is the wing’s mass moment of inertia with respect to the fixed point of rotation *O* and $\lambda_{k}$ is a constant defined by
(11)$$\lambda_{k}=\int_{m_{w}}y\phi_{k}\;dm_{w}$$

The potential energy of the wing $U_{wing}$ is
(12)$$\begin{array}{c} U_{wing}=\int_{V}S\left(W,W\right)\;dV \end{array}$$
where S is a symmetric, quadratic strain energy density function and *V* is the wing’s volume. We assume both wings to flap and deform symmetrically.

With potential and kinetic energies defined, we apply Lagrange’s equation to arrive at the equations of motion governing thorax displacement *x* and wing modal response $q_{k}$ as
(13)$$\begin{array}{ccc} \left(m+\frac{2I_{0}}{\Gamma^{2}}\right)\ddot{x}+\frac{2}{\Gamma }\sum_{k=1}^{\infty }\lambda_{k}{\ddot{q}}_{k}+kx & = & F_{NC} \end{array}$$
(14)$$\begin{array}{ccc} \frac{\lambda_{k}}{\Gamma }\ddot{x}+{\ddot{q}}_{k}+\left(\omega_{k}^{2}-\frac{{\dot{x}}^{2}}{\Gamma^{2}}\right)q_{k} & = & Q_{NC},k \end{array}$$
where $F_{NC}$ are the non-conservative physical forces applied to the thorax, $Q_{NC,k}$ is the non-conservative modal force exciting the wing’s *k*th vibration mode, and $\omega_{k}$ is the wing’s *k*th natural frequency.

### 2.3. Non-Conservative Forces

We next consider the non-conservative forces acting on the system. Non-conservative forces include those applied by the indirect flight muscle to the thorax, denoted F(t), and aerodynamic forces acting over the wing surfaces, denoted $F_{aero}\left(y,t\right)$. Using the blade-element approach from [20], we estimate $F_{aero}\left(y,t\right)$ as
(15)$$F_{aero}\left(y,t\right)=-\frac{1}{2}C_{D}\rho_{f}\int_{S}\left({\dot{\theta }}^{2}y^{2}+2\dot{\theta }\dot{W}y\right)dS+O\left(W^{2}\right)$$
where dS is the differential surface over which the aerodynamic force acts, $C_{D}$ is a drag coefficient, and $\rho_{f}$ is fluid density. The terms of order $W^{2}$ are neglected since wing deformation is assumed to be small. Due to the SDOF rotation of the wing, the only aerodynamic force acting on the wing is drag (from a quasi-static perspective).

Using the principle of virtual work, we can identify the total non-conservative forces acting on the thorax as
(16)$$\begin{array}{c} F_{NC}=F\left(t\right)-C_{D}\rho_{f}\frac{{\dot{x}}^{2}}{\Gamma^{3}}\int_{S}y^{3}dS \end{array}$$
where the first term represents muscle forces acting directly on the thorax, and the second term represents the propagation of aerodynamic forces acting on the wing to the thorax. The non-conservative modal forces acting to deform the wing are
(17)$$Q_{NC,k}=-C_{D}\rho_{f}\frac{{\dot{x}}^{2}}{\Gamma^{2}}\int_{S}y^{2}\phi_{k}dS-2C_{D}\rho_{f}\frac{\dot{x}}{\Gamma }{\dot{q}}_{k}\sum_{r=1}^{\infty }\int_{S}y\phi_{k}\phi_{r}dS$$

Above, the first term is an aerodynamic loading term dependent only on the angular velocity of the wing’s rigid body rotation. The second term is an aerodynamic damping term that is dependent both on the wing’s rigid body angular velocity as well as the rate of deformation.

Lastly, we derive expressions for the instantaneous power and energy expended by the flapping wing system. These expressions are useful when quantifying the influence of wing flexibility on the energetic economy of flight. The instantaneous power $P_{F}\left(t\right)$ delivered to the thorax by the non-conservative applied force F(t) is
(18)$$P_{F}\left(t\right)=F\left(t\right)\dot{x}$$
and the energy utilized by the applied force over a period of time (e.g., the energy input into the system) is
(19)$$E_{in}=\int \left|P_{F}\left(t\right)\right|dt$$

## 3. Results

### 3.1. Simulation Parameters

We numerically simulate Equations (Equation 13) and (Equation 14) to determine the dynamic responses of the thorax and wing. The simulation parameters are summarized in Table 1. Parameters are estimated for the hawkmoth *Manduca sexta*, a common model organism in the field of flapping wing flight.

Wing mass, length and width are approximated from [21]. Wing width is assumed to be constant, and mass is assumed to be uniformly distributed. The wing’s moment of inertia about point of rotation *O* can then be calculated via $I_{O}=\frac{1}{3}m_{W}L^{2}$. Thorax stiffness is estimated from [9] and thorax mass from [22]. The transmission ratio is idealized from [23], which reports a modestly nonlinear relationship between thorax compression and wing angle. The muscle forces acting on the thorax are assumed to be harmonic and of the form $F\left(t\right)=F_{0}\sin\left(2\pi ft\right)$, where $F_{0}$ is the force amplitude and *f* is the forcing frequency, both idealized from [9]. The fluid density and drag coefficient are taken from [20]. Note that the drag coefficient generally varies with respect to the wing’s angle of attack. However, because the wing experiences only SDOF rotation and there is no free-stream velocity, the angle of attack is always $\pm \frac{\pi }{2}$. The drag coefficient can thus be treated as a constant.

We retain vibration modes with natural frequencies less than 10 times that of the flapping frequency, as modes with higher natural frequencies will not be excited appreciably. For this simulation, only one mode (first bending mode) is retained. We determine the first mode shape and natural frequency using a user-defined finite element model (see Appendix A). The wing’s natural frequency depends on its density, geometric properties and Young’s modulus. We tune the Young’s modulus to a value of 9.5 GPa such that the wing’s first natural frequency agrees with the first measured natural frequency of the hawkmoth forewing [24]. This modulus value is within the reported range for cuticles in the hawkmoth thorax [25].

The wing and thorax responses are determined numerically using Matlab’s (Ver. R2020b) ode45 solver. We calculate responses over 50 total wingbeats and consider 100 evenly spaced time steps per wingbeat. All initial conditions are zero. Reported data are shown for the steady-state response of the system. To better assess the influence of wing flexibility on system dynamics, we simulate a rigid wing system (RWS) to compare against a flexible wing system (FWS). For reference, the natural frequency of the RWS is about 23.5 Hz, slightly higher than the 21.9 Hz first natural frequency of the FWS.

### 3.2. Baseline Response

We first simulate a system with parameters idealized from those of a hawkmoth (Table 1) to verify that the responses are in proximity to those measured in biology. Given the model assumptions (e.g., SDOF wing rotation and linear thorax stiffness), we do not expect the model dynamics to emulate insect mechanics perfectly. However, responses are expected to be on the same order of magnitude so that the model has physical relevance.

Thorax displacement, wingtip displacement and aerodynamic force are shown in Figure 2. The thorax displacement amplitude is about 0.5–0.6 mm for the RWS and FWS, which is similar to the maximum deformation measured on the dorsal surface of the hawkmoth *Agrius convolvuli* thorax during tethered flight [26]. To our knowledge, thorax deformation has not been reported in *M. sexta*, but the morphological similarity between the two species makes this a suitable comparison.

It is more challenging to compare wingtip displacements estimated by the model to those measured experimentally in flapping insects. Rigid body rotational kinematics are generally estimated by tracking the three-dimensional trajectory of points on a wing surface. However, if the wing deforms at any of the measurement points, the reconstructed kinematics will be influenced by both rotation and deformation. It is consequently challenging to decouple out-of-plane elastic deformation from rigid body rotation in free flying insects. Nonetheless, we can approximate deformation based on available data. Willmott and Ellington showed that the angle of rotation (defined as the angle between the horizontal and the wing’s trailing edge) varies considerably between the proximal and distal portion of the wing [27]. Assuming that the angle of rotation of the proximal portion of the wing represents rigid body rotation, the increased angle of rotation observed at the more distal portion of the wing must arise from deformation at the wing’s trailing edge. If the distal portion of the wing rotates about 30 degrees beyond the rigid body rotation, and we assume a chord width of 15 mm, the trailing edge of the wing would deform about 8.7–26 mm. Our model predicts a wingtip deflection of about 10 mm, which falls within this estimated range.

The peak aerodynamic drag produced by both the RWS and FWS is about 35 mN and averages to zero over a single wingbeat. The peak drag predicted by this model will be similar to the magnitude of the aerodynamic force vector (inclusive of both lift and drag) of a moth wing experiencing more realistic wing kinematics. Numerical simulations of hovering *M. sexta* indicate that a rigid wing produces peak forces of about 25 mN in the insect’s dorsal–ventral and anterior–posterior axes and about 10 mN in the lateral axis [28]. The peak forces in the dorsal–ventral and anterior–posterior axes occur at the same moment at which the instantaneous force vector has a magnitude of 35 mN. Based on the thorax deflection, wingtip deflection and aerodynamic force, the simplified model derived here appears to capture several dynamic features of the real insect.

We now compare the responses of the RWS and FWS. The FWS thorax deforms 20% less than the RWS thorax. The FWS wing consequently experiences rotation and angular velocity amplitudes 20% lower compared to the RWS wing as well. Despite having a considerably lower angular velocity, the FWS produces peak aerodynamic forces only 4% lower than the RWS. This implies that, within the FWS, the wing’s elastic deformation contributes non-trivially to the overall aerodynamic force production. For the FWS, the peak forces due to rigid body rotation are about 27 mN, compared to 11 mN due to elastic deformation. Since both the FWS and RWS are forced post-resonance, the phase of thorax deformation lags the applied muscle by about $\frac{\pi }{2}$. In the FWS, wing deformation lags thorax displacement by about $\frac{\pi }{4}$, which suggests that the aerodynamic and inertial forces causing wing deformation are similar in magnitude. The inertial forces acting to deform the wing are proportional to the wing’s rigid body angular acceleration. If inertial forces dominate aerodynamic forces, the phase between thorax displacement and wing deflection will be close to zero. The aerodynamic forces acting to deform the wing are proportional to the rigid body angular velocity. If aerodynamic forces dominate inertial forces, the phase between the thorax displacement and the wing deflection will be about $\frac{\pi }{2}$. Consequently, when aerodynamic and inertial forces are similar in magnitude, the resulting phase lag between thorax deformation and wing deformation approaches $\frac{\pi }{4}$. Forces associated with wing deformation propagate back to the thorax in the FWS, which is why the thorax response in the FWS slightly lags the thorax response of the RWS.

### 3.3. Energetics of the Baseline System

Next, we consider how wing flexibility influences the energetic requirements of flapping. The power delivered to the thorax is shown over two wingbeats for the RWS and FWS in Figure 3. Power requirements for both systems are similar in magnitude to reported estimates for *M. sexta* [17,21]. The peak and mean power required for the FWS are about 14% and 21% less, respectively, than what is required by the RWS. The lower power requirement stems from the difference in peak thorax velocities, which is about 14% lower in the FWS. The input energy to the FWS is 2.19 mJ, compared to 2.73 mJ for the RWS; thus, the FWS requires 20% less total energy compared to the RWS. Considering that aerodynamic force generation is similar between the two systems, these results suggest that wing flexibility contributes considerably to efficiency in flapping flight.

The potential energy stored within the FWS and RWS during flapping is shown in Figure 3. The RWS stores potential energy only in the thorax, whereas the FWS stores potential energy both in the wing (about 40% of total potential energy storage) and thorax (about 60% of total potential energy storage). The RWS has greater potential energy storage with a maximum of about 0.5 mJ, while the FWS stores maximally 0.45 mJ. The wing and thorax store potential energy at different phases of the wingbeat in the FWS, and consequently the potential energy does not reach zero over a wingbeat. The lower bound of potential energy in the FWS is about 0.1 mJ. Thus, the RWS recovers all 0.5 mJ while the FWS recovers only 0.35 mJ. However, as previously mentioned, the FWS requires 20% less input energy relative to the RWS. Potential energy recovery is therefore not individually a good indicator of the energetic benefits conferred by wing flexibility.

### 3.4. Parameter Studies

The model predicts responses (thorax deformation, wingtip displacement, etc.) in proximity to what has been reported for the hawkmoth, and thus may serve as a useful tool when investigating the sensitivity of the system response to changing parameters. Here, we use our model to examine how thorax displacement, aerodynamic force, and energy input vary with force amplitude $F_{0}$, transmission ratio Γ, thorax stiffness *k*, wing mass $m_{w}$ and flapping frequency *f*. We report the mean rectified aerodynamic force, since the aerodynamic force itself will average to zero over a single wingbeat. We also quantify the energetic benefit of wing flexibility by taking the ratio of the energy input to the FWS to the energy input of the RWS (energy ratio hereafter). When the energy ratio is less than one, the FWS is more energetically economical than the RWS.

#### 3.4.1. Force Amplitude

First, we determine how force amplitude influences the system dynamics. The equation of motion governing the thorax response Equation (Equation 13) and the wing’s elastic response Equation (Equation 14) have quadratic damping terms, meaning the system damping increases at higher response amplitudes. Peak frequencies decrease with increasing damping ratios in linear systems, so it is plausible that peak responses will occur at lower frequencies for higher force amplitudes (and consequently, increased damping) for the system considered here. We consider force amplitudes of 1.0 N, 1.5 N and 2.0 N across a flapping frequency range of 15–35 Hz. The system responses are shown in Figure 4.

Variable force amplitude influences the system performance considerably. With increasing force amplitude, the flap frequency that corresponds to the largest thorax displacement decreases in both the FWS and RWS due to the increase in aerodynamic damping. The peak flap frequency in the RWS is higher for all force amplitudes because the linear natural frequency of the RWS is greater than that of the FWS. Interestingly, the greatest mean rectified aerodynamic force occurs at a higher flapping frequency than the peak thorax displacement, which indicates that maximal thorax displacement does not correspond to maximal aerodynamic force production. Instead, the greatest mean aerodynamic force occurs at a flap frequency that corresponds to the highest thorax velocity. The idea that maximum thorax displacement and velocity occur at different peak frequencies is consistent with the results of [19], which show that multiple peak frequencies exist in a linear flapping-wing system depending on the specific transfer function (e.g., force-to-displacement and force-to-velocity) considered. Similar to thorax displacement, the flap frequency corresponding to maximum energy input tends to increase with increasing force amplitude. However, the energy input in general increases non-linearly with force amplitude; a maximum energy input for both the RWS and FWS is about 1.8 mJ for a force amplitude of 1 N and nearly 5 mJ for a force amplitude of 2 N. Energy input is comparable between the RWS and FWS at lower flap frequencies, whereas the energy input to the FWS is lower at higher flap frequencies. As force amplitude increases, the energy ratio between the FWS and RWS tends to favor the RWS at increasing flap frequency ranges. This indicates that the FWS outperforms the RWS energetically at higher forces.

#### 3.4.2. Transmission Ratio

Next, we evaluate how varying transmission ratio Γ affects the system. The transmission ratio influences the effective mass of the thorax Equation (Equation 13) due to the rigid coupling between the thorax and wing, where the effective mass and transmission ratio are inversely proportional. The aerodynamic forces are back-propagated to the thorax from the wing scale with the inverse of the the transmission ratio cubed, and the dominant aerodynamic forces acting to deform the wing scale with the inverse of transmission ratio squared. We consider transmission ratios of 0.6 mm/rad, 0.8 mm/rad N and 1.0 mm/rad across a flapping frequency range of 15–35 Hz. The system responses are shown in Figure 5.

Thorax displacement in both the RWS and FWS as well as the flapping frequency at which peak thorax displacement occurs both grow as the transmission ratio increases and effective mass decreases consequentially. Each trend is described by this decrease in effective mass. As the effective mass is lowered, the linear natural frequency of both the RWS and FWS increases. Similarly, because there is less effective mass to resist input forces, the thorax amplitude is greater. The increased motion of the thorax at higher transmission ratios (and higher thorax velocity) causes greater mean rectified aerodynamic forces and maximal energy inputs for both systems. The most dramatic effect of the variable transmission ratio occurs with the input energy ratio of the FWS and RWS. For all transmission ratios, RWS and FWS energy inputs grow similarly with flapping frequency until reaching a maximum; beyond this maximum, the energy input of the FWS rolls off more quickly. Consequently, at higher frequencies, the energy ratio favors the FWS. The energy ratio favors the FWS for lower transmission ratios because the flapping frequency corresponding to maximum energy input is lower for reduced transmission ratios. It is worth noting that the aerodynamic force rolls off as a function of flap frequency quicker in the FWS as well. However, because the maxima of aerodynamic force and energy input occur at different frequencies, there is a narrow frequency band in which the FWS has lower energetic expenditures and comparable aerodynamic force generation (e.g., from about 20–25 Hz when Γ = 0.8 mm/rad).

#### 3.4.3. Thorax Stiffness

Thorax stiffness *k* may also affect the performance of the system. Similar to the transmission ratio, adjusting the thorax stiffness will influence the system’s linearized natural frequencies. However, unlike the transmission ratio, the thorax stiffness does not explicitly affect the aerodynamic forces; aerodynamic forces are only implicitly affected by a change in the thorax response velocity. We consider thorax stiffness values 2280 N/m, 2850 N/m, and 3420 N/m across a flapping frequency range of 15–35 Hz. The system responses are shown in Figure 6.

As the thorax becomes stiffer, the maximum thorax displacement decreases, and the peak frequency of thorax displacement increases. However, the peak achievable aerodynamic force is largely preserved across all values of thorax stiffness considered. This implies that despite a reduction in thorax displacement amplitude, a stiffer thorax can achieve the same velocity as a more compliant thorax if excited at a higher frequency. Interestingly, the peak energy requirements decrease as the thorax becomes stiffer in both the RWS and FWS. Because the aerodynamic force generation is similar across the values of *k* tested, this suggests that a stiffer thorax may be energetically favorable. Similar to the trends observed in other parametric studies, the energy ratio favors the FWS at mid-to-high flapping frequencies.

#### 3.4.4. Wing Mass

Lastly, we explore the effect of wing mass on system performance. Wing mass affects the wing’s moment of inertia, and hence the system’s natural frequencies. In the case of the FWS, wing mass also adjusts the relative magnitude of the forces acting to deform the wing. Lightweight wings with low inertia will deform primarily under aerodynamic loading, whereas heavier wings will deform primarily from inertial forces. Consequently, wing mass is expected to influence the system-level dynamics. We consider wing masses of 35 mg, 50 mg, and 65 mg across a flapping frequency range of 15–35 Hz. The system responses are shown in Figure 7.

Heavier wings tend to cause larger thorax displacements due to the increased inertia in the system. Because the system’s linear natural frequencies decrease with increasing wing mass, the flap frequency corresponding to maximal thorax displacement decreases as well. The maximum achievable aerodynamic force is relatively insensitive to wing mass, though the flap frequency at which the maximum aerodynamic force occurs reduces as the wing mass increases. Similar to increasing the transmission ratio, energy input increases with wing mass as well due to the increased inertia in the system. Lightweight wings are thus energetically favorable. For all wing masses considered, the energy ratio decreases monotonically with respect to flap frequency.

## 4. Discussion

Flapping wing insects often serve as inspiration in the design of small-scale FWMAVs owing to their small size and excellent energy efficiency. Though models have been developed to investigate the dynamics of flapping insects, many of these models neglect wing flexibility, which has been shown in isolated wing models to have a large effect on aerodynamic and energetic performance. In this work, we derive a simplified system-level model of the insect wing system considering the thorax coupled to a flexible wing and surrounding fluid environment. The model is applied to estimate the dynamics of a hawkmoth. Despite its simplifications, the model predicts dynamic responses such as thorax deformation and aerodynamic force generation in proximity to biological measurements in the same insect. When using model parameters similar to those of a hawkmoth, wing flexibility reduces energy expenditures by about 20% compared to an equivalent system with rigid wings. These energetic savings conferred by the flexible wing are relatively insensitive to variations in model parameters.

### 4.1. System Energetics

In general, wing flexibility reduces the energetic costs of flapping. Within the RWS, the input energy is (1) stored as potential energy in the thorax, (2) stored as kinetic energy in the thorax and wing inertia, and (3) dissipated via aerodynamic forces. In the FWS, a portion of input energy is directed towards the potential and kinetic energy associated with wing deformation and deformation rate, respectively. The energy redistribution within the FWS results in a lower thorax velocity, and hence a reduced peak and mean power compared against the RWS. The lower thorax velocity in the FWS implies that the wing’s experience a lesser rigid body angular velocity. However, because the wing is free to deform in the FWS, the wingtip velocity is similar between the FWS and RWS. The aerodynamic performance is thus comparable between the two systems, despite the FWS having a lower flapping amplitude. Studies show the energetic savings of the FWS are largely insensitive to deviations from the baseline parameters, and that there are flapping frequency bands over which the FWS outperforms the RWS while maintaining similar aerodynamic force generation.

We hypothesize that the energetic benefits offered by flexible wings could be realized in a system with rigid wings if a series elastic element were placed between the thorax and wing. Recent studies suggest that elasticity in the insect drivetrain may be modeled using both a parallel element (such as the one considered in this work) and a series element realized as a torsional spring at the wing’s point of rotation [9]. Inclusion of a series element would allow the wing to passively rotate beyond what is prescribed by thorax deformation, thereby converting the transmission between thorax deformation and wing rotation from a static mapping to a dynamic mapping. The wing could consequently achieve larger angular rotations and velocities at lower thorax velocities, thereby reducing the instantaneous power required of the applied force and energy input. This is similar to a system with flexible wings and a static transmission because the wing deformation increases the effective angle of rotation between the base and tip of the wing, though the angle at the wing base is fully defined by the thorax position. It is possible that additional energetic benefits may be realized if a series element combined with a flexible wing; however, it requires additional investigation to understand how these two components would function together.

This research also substantiates an assumption employed in flapping wing studies. When calculating the power required to flap a wing (flexible or rigid), it is sometimes assumed that the negative power offsets the positive power requirements, or that negative work is stored by an elastic structure and later recovered [17,29]. However, the upstream energy-storing element is not generally modeled. By explicitly modeling the thorax, we can remove this assumption and directly assess the power delivered to the thorax by the applied force rather than looking at the power at the wing base. The negative power of the non-conservative force is no longer recoverable and is instead a result of the power required to deccelerate the thorax motion. In fact, because braking can be achieved via passive elastic elements in the system, negative power represents a system inefficiency. Thus, by investigating the coupled wing–thorax system, we are able to relax assumptions regarding energy recovery within the system.

### 4.2. Model Limitations

While this simple model provides insights into the dynamics of a flapping wing system, it has limitations that should be considered in application or future modeling efforts. First, the model assumes that the wings experience only SDOF rotation and consequently do not produce lift. From a quasi-static perspective, the aerodynamic force modeled in this work is analogous to drag. To produce lift in a quiescent environment, the wings must experience at least two degrees of rotation (roll and pitch). Under this condition, the angle-of-attack will vary over the wing-beat cycle and lift will be non-zero. Both lift and drag forces would be assumed to act at the wing quarter chord such that the wing experiences both bending and twisting type deformations (as opposed to the current model which accommodates only bending). Under this case, bending and twisting affect the wing shape and hence the aerodynamic forces generated. Future modeling efforts will consider the fluid–structure interaction of flapping wings with more realistic flapping kinematics.

Next, the mechanics of the flapping insect wing hinge are not fully understood and require further investigation. As discussed previously, it is possible that the transmission between thorax and wing is dynamic and contains a series-elastic element, though series elastic effects appear to be variable between insect species [19]. Additional research is required to characterize the behavior of the wing hinge so that it can be incorporated into mathematical models.

Lastly, the thorax structure is complex and also not well-understood. Force–displacement testing has shown that the thorax may behave as a nonlinear hardening spring at large displacements, and that damping within the thorax may better be described by a viscous damping model rather than a structural damping model [9]. However, it is uncertain whether these thorax characteristics generalize across insect species; thus, treating the thorax as a simple sprung mass is a suitable starting place when modeling the dynamics of the flapping system.

### 4.3. Design Guidelines

The simple model derived in this work can begin to help provide design insights into insect-inspired FWMAVs. At centimeter- or millimeter-length scales, FWMAVs tend to be driven by piezoelectric, dieletric/electrostatic elastomers, electromagnetic or other reciprocating actuators [30]. In any case, the stiffness of the actuator itself best parallels the stiffness of the insect thorax. Small actuator displacements may be transformed into wing rotation by a mechanical transmission, which behaves somewhat like the insect wing hinge.

Our model shows that the system’s natural frequency is governed primarily by the actuator stiffness, the mechanical advantage of the wing-thorax transmission and the rotational inertia of the wings. Though actuator mass and wing compliance also affect the system’s first natural frequency, the influence of these components is small. Actuator stiffness, wing inertia and transmission ratio should be selected such that the system operates slightly above the the system’s force-to-actuator-displacement resonant frequency, since this represents an aerodynamically and energetically favorable configuration. Given constraints on actuator and wing design and/or selection (available manufacturing processes, off-the-shelf-components, etc.), it is perhaps most practical to adjust the transmission ratio to modify the natural frequency. The system natural frequency scales with the inverse of the transmission ratio squared and is thus sensitive to this parameter.

Once an initial design is available, wing compliance should be considered in order to enhance vehicle aerodynamic force generation and energetic economy. Our results show a considerable frequency band over which flexible wings produce comparable aerodynamic forces for lower-input energy relative to rigid wings (Figure 4, Figure 5, Figure 6 and Figure 7). Previous studies show that many insects flap at about $\frac{1}{5}$ to $\frac{1}{2}$ of the first natural frequency of their wings [31], and computational studies indicate flapping at about 1/3 the wing’s natural frequency is aerodynamically advantageous [32]. Within this range is a reasonable target for FWMAV wings. Practically, artificial wing natural frequencies can be tuned in a variety of ways, such as adjusting the layout and effective diameter of vein networks, material selection, or adjusting tension of the membrane support between wings. The performance of the system can then be characterized by quantifying energy input and aerodynamic force as a function of the flapping frequency.

Continued efforts in studying and modeling flapping wing insects will bolster novel designs of insect-inspired FWMAVs. A comprehensive understanding of how these insects are able to perform efficiently in flight is vital for these emerging technologies to become practicable.

## Figures and Tables

**Figure 1 biomimetics-07-00207-f001:**
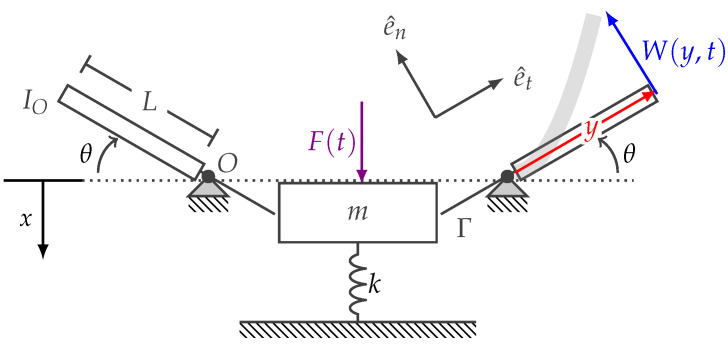
Simplified dynamic model of a flapping wing insect.

**Figure 2 biomimetics-07-00207-f002:**
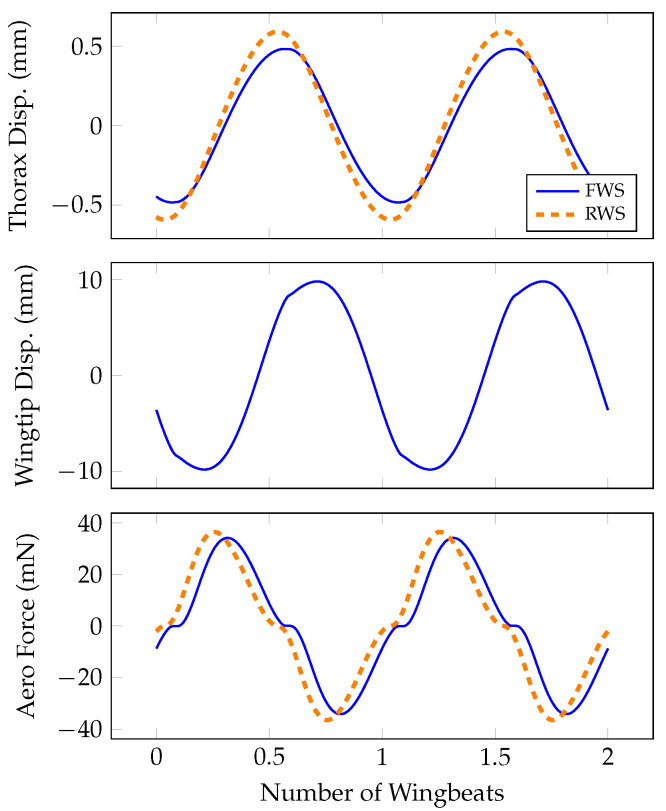
Thorax displacement *x*, wingtip deformation W(L,t) and aerodynamic force $F_{aero}$ over two wingbeats.

**Figure 3 biomimetics-07-00207-f003:**
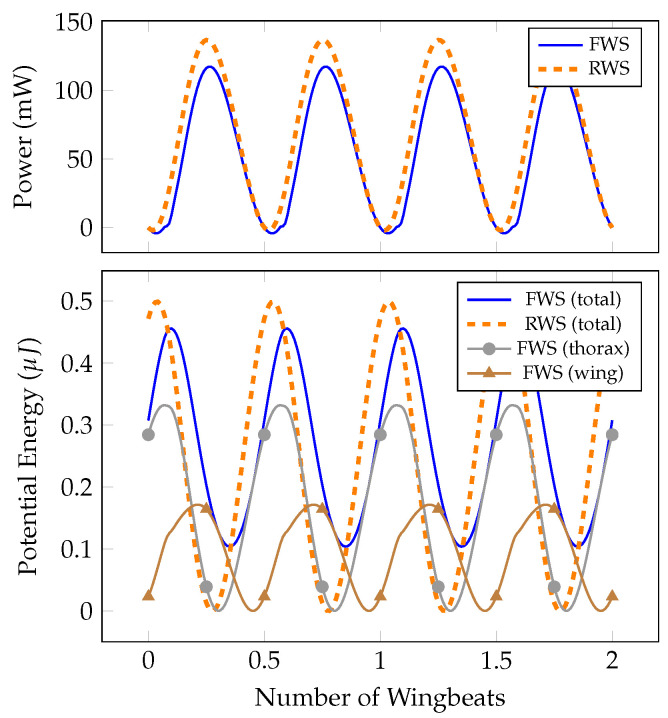
Power delivered to the thorax by applied force F(t) and potential energy storage for RWS and FWS.

**Figure 4 biomimetics-07-00207-f004:**
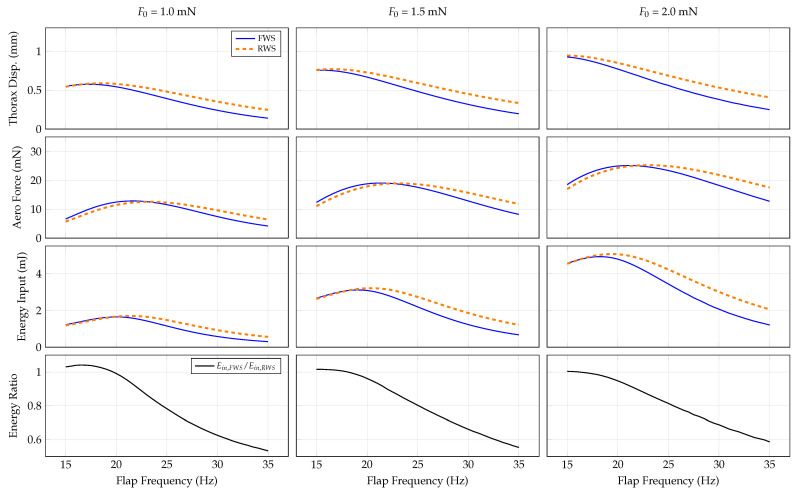
System response as a function of flap frequency and force amplitude for the RWS and FWS.

**Figure 5 biomimetics-07-00207-f005:**
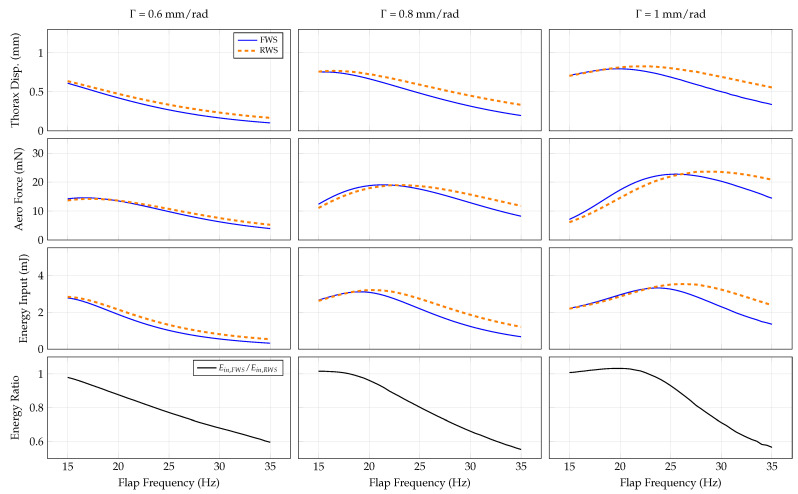
System response as a function of flap frequency and transmission ratio for the RWS and FWS.

**Figure 6 biomimetics-07-00207-f006:**
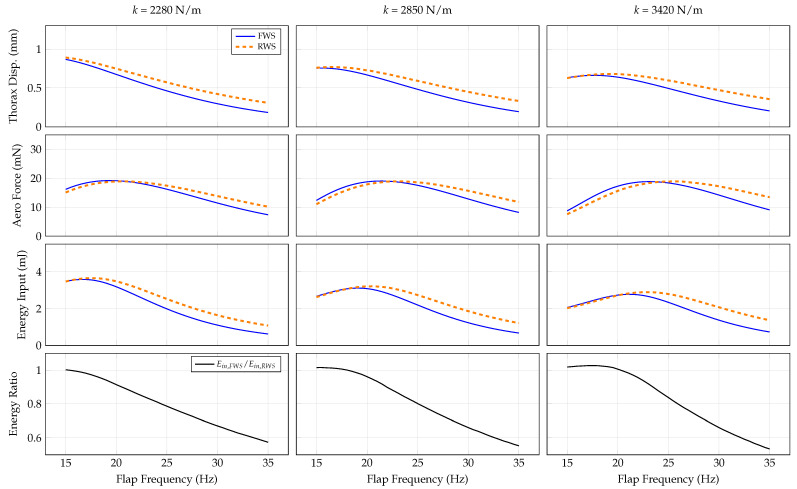
System response as a function of flap frequency and thorax stiffness for the RWS and FWS.

**Figure 7 biomimetics-07-00207-f007:**
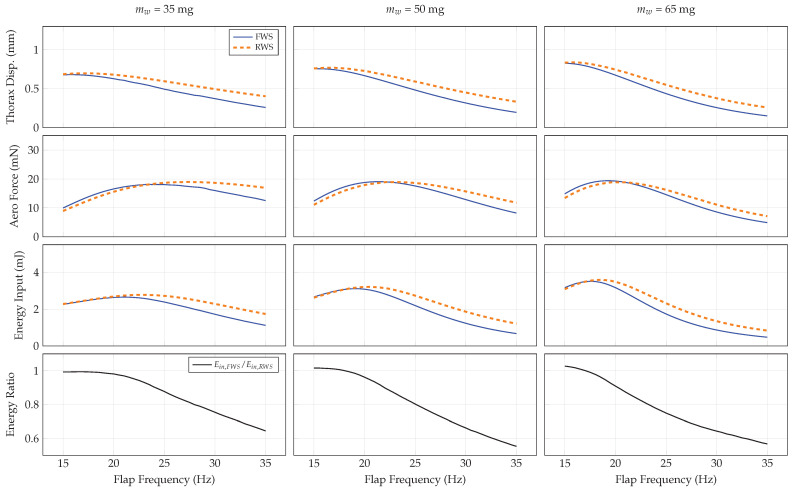
System response as a function of flap frequency and wing mass for the RWS and FWS.

**Table 1 biomimetics-07-00207-t001:** Numerical simulation parameters. Parameters are representative of a hawkmoth *M. sexta*.

Parameter	Symbol	Value	Unit
Wing mass	$m_{w}$	50	mg
Wing length	*L*	5	cm
Wing width	*w*	1.8	cm
Wing thickness	$t_{w}$	50	μm
Wing moment of inertia	$I_{0}$	0.42	g-cm^2^
Wing natural frequency	$\omega_{1}$	60	Hz
Thorax mass	*m*	0.7	g
Thorax stiffness	*k*	2850	N/m
Transmission Ratio	Γ	0.80	mm/rad
Forcing Amplitude	$F_{0}$	1.5	N
Forcing Frequency	*f*	25	Hz
Drag Coefficient	$C_{D}$	3	-
Fluid Density	$\rho_{f}$	1.25	kg/m^3^

## Data Availability

Models are available by the corresponding author upon reasonable request.

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
