# Peer review of "Modeling and Analysis of a Simple Flexible Wing—Thorax System in Flapping-Wing Insects"

_biomimetics, 2022, doi:10.3390/biomimetics7040207_

Round 1
Reviewer 1 Report
In this manuscript, Cote et al. developed a numerical dynamic model to investigate the effects of flexibility of the wings on the energetics and performance of a FWMAV. This paper is well written and the novelty is clear. The reviewer can recommend for publication with the following comments.
The reviewer has some concerns on the assumption of fixed angle of attack in this study. The authors stated that this model is SDOF and the angle of attack is fixed at pi/2. However, the authors are suggested to add explanations on how this assumption can represent the real situations where the angle of attack is varying during a wingbeat (i.e. the reliability of this simplified model).
Line 455, ‘dialetric’ should be ‘dielectric’.
Author Response
Thank you for your review of our manuscript. We appreciate your comments and believe they have improved the quality of our work.
We agree that the SDOF and fixed angle-of-attack assumptions limit the generality of our model. However, as mentioned in the introduction, simplified SDOF models have been used in the literature several times in recent years to study the phenomena associated with flapping wing flight in a simplified environment. For example, Lynch et al. used an SDOF flapping wing model and robophysical experiment to demonstrate that the effective resonant frequency of a thorax/rigid wing system is influenced by fluid damping. Pons and Beatus used an SDOF model to show different resonant states in a flapping wing system. Here, we use a modified SDOF model to demonstrate how wing flexibility can influence system-level dynamics. This simple model provides foundational knowledge as we work to develop a more realistic model that accommodates MDOF rotation and a time-varying angle-of-attack.
We have added the following language in the introduction and discussion regarding this assumption and how the methodology derived within may be extended to MDOF flapping situations. The changes are highlighted in the revised manuscript.
Intro (final paragraph):
Non-lifting models are simple yet useful tools for investigating the physics of flapping flight and are common in precedent work. Insights garnered from this model can inform the compliant design of bioinspired FWMAVs and lay a foundation for more complex models that accommodate MDOF rotations and lifting flows.
Discussion (model limitations):
First, the model assumes that the wings experience only SDOF rotation and consequently do not produce lift. From a quasi-static perspective, the aerodynamic force modeled in this work is analogous to drag. To produce lift in a quiescent environment, the wings must experience at least two degrees of rotation (roll and pitch). Under this condition, the angle-of-attack will vary over the wing-beat cycle and lift will be non-zero. Both lift and drag forces could be assumed to act at the wing quarter chord such that the wing experiences both bending and twisting type deformations (as opposed to the current model which accommodates only bending). Under this case, bending and twisting affect the wing shape and hence the aerodynamic forces generated. Future modeling efforts will consider the fluid-structure interaction of flapping wings with more realistic flapping kinematics.
Additionally, we have corrected the spelling error ‘dielectric’. Thank you again for reviewing our manuscript, and we hope that we have addressed your concerns adequately.
Reviewer 2 Report
The manuscript describes an idealized model of a flapping-wing insect based on Lagrangian dynamics. Simulations and parameter studies are included. The model is sound and the manuscript is well written.

Author Response
Thank you for your review of our manuscript. We appreciate your comments and believe they have improved the quality of our work. We have gone through and updated our manuscript per your edits, including clarifying equations, standardizing the bibliography capitalization/scientific names conventions, and addressing grammatical mistakes. Thank you again for reviewing our manuscript, and we hope that we have addressed your concerns adequately.